# Geographic Inequalities of Respiratory Health Services Utilization during Childhood in Edmonton and Calgary, Canada: A Tale of Two Cities

**DOI:** 10.3390/ijerph17238973

**Published:** 2020-12-02

**Authors:** Jesus Serrano-Lomelin, Charlene C. Nielsen, Anne Hicks, Susan Crawford, Jeffrey A. Bakal, Maria B. Ospina

**Affiliations:** 1Department of Obstetrics and Gynecology, Faculty of Medicine and Dentistry, University of Alberta, Edmonton, AB T6G 2S2, Canada; jaserran@ualberta.ca; 2School of Public Health, University of Alberta, Edmonton, AB T6G 1C9, Canada; ccn@ualberta.ca; 3Department of Pediatrics, Faculty of Medicine and Dentistry, University of Alberta, Edmonton, AB T6G 1C9, Canada; eagreene@ualberta.ca; 4Alberta Perinatal Health Program, Alberta Health Services, Calgary, AB T2N 2T9, Canada; Susan.Crawford@albertahealthservices.ca; 5Provincial Research Data Services, Alberta Health Services, Edmonton, AB T6G 2C8, Canada; jbakal@ualberta.ca

**Keywords:** health inequalities, spatial filters, early childhood, respiratory diseases

## Abstract

Young children are susceptible to respiratory diseases. Inequalities exist across socioeconomic groups for paediatric respiratory health services utilization in Alberta. However, the geographic distribution of those inequalities has not been fully explored. The aim of this study was to identify geographic inequalities in respiratory health services utilization in early childhood in Calgary and Edmonton, two major urban centres in Western Canada. We conducted a geographic analysis of data from a retrospective cohort of all singleton live births occurred between 2005 and 2010. We aggregated at area-level the total number of episodes of respiratory care (hospitalizations and emergency department visits) that occurred during the first five years of life for bronchiolitis, pneumonia, lower/upper respiratory tract infections, influenza, and asthma-wheezing. We used spatial filters to identify geographic inequalities in the prevalence of acute paediatric respiratory health services utilization in Calgary and Edmonton. The average health gap between areas with the highest and the lowest prevalence of respiratory health services utilization was 1.5-fold in Calgary and 1.4-fold in Edmonton. Geographic inequalities were not completely explained by the spatial distribution of socioeconomic status, suggesting that other unmeasured factors at the neighbourhood level may explain local variability in the use of acute respiratory health services in early childhood.

## 1. Introduction

Paediatric respiratory diseases are a leading cause of morbidity in childhood, particularly among children of preschool age [1]. Early-childhood respiratory diseases have long-term negative consequences on adult health with major economic consequences for health systems [1]. Pneumonia, bronchiolitis, tuberculosis, and asthma are among the most common respiratory diseases affecting young children [2]. Individual factors such as preterm birth, maternal smoking, and maternal asthma, among others, have been related to lung function impairment in early life [3].

Socioeconomic factors have been identified as major drivers of respiratory diseases [4]. Respiratory diseases are generally more frequent among children from lower socioeconomic status (SES) compared to those born in high-SES families [5,6,7]. The relationship between SES and respiratory health has been partially attributed to individual factors associated with poverty (e.g., undernutrition and adverse living conditions) and other structural determinants of health [7,8,9].

In Canada, respiratory health inequalities across socioeconomic groups have been identified for a variety of respiratory outcomes in adult populations, including smoking [10] and chronic obstructive pulmonary disease [11,12,13]. Inequalities in paediatric respiratory health have been described for asthma hospitalizations across the urban/rural divide [14], asthma emergency department visits across (large) health zones in Alberta [15], respiratory infections among First Nations and Inuit children [16], and differences in influenza hospitalizations between First Nations populations living on and off reserves [17]. Recently, Belon et al. [18] described paediatric respiratory inequalities across socioeconomic groups for a variety of respiratory diseases in Alberta, calling for more research examining variations at small-scale levels, for example, within urban areas and among rural/remote areas. Both urban and rural areas have experienced different levels of population growth during the last decade in Alberta [19], likely impacting the development and access to health care facilities.

The geography of health inequalities looks over the role of place of residence in shaping health gaps among populations [20,21]. It determines the spatial extension of health inequalities while linking contextual factors (i.e., social and environmental features embedded into a place), and promoting further investigation of relevant social, environmental, and political questions [22]. To our knowledge, there is no evidence on geographic differences in the utilization of paediatric respiratory health services in urban centres in Alberta. This study aimed to identify geographic inequalities in the prevalence of respiratory health services utilization in early childhood in two major urban centres in Alberta and quantify the magnitude of spatial gaps in respiratory health services use. The study is exploratory in nature given the lack of past evidence on geographic inequalities in the utilization of paediatric respiratory health services in both cities. Results from this study can guide future research on the contextual factors that may produce intraurban inequalities and provide a basis to support local actions aimed at reducing them.

## 2. Materials and Methods

### 2.1. Study Design and Setting

This study analyses data on events of hospitalizations and emergency department visits due to respiratory outcomes from a retrospective cohort of all singleton live births (≥22 weeks of gestation) that occurred in Alberta between 1 April 2005 and 31 March 2010. Ethics approval for the study was obtained from the University of Alberta’s Health Research Ethics Board (Pro00088569). The study is reported as per the STrengthening the Reporting of OBservational studies in Epidemiology (STROBE) guidelines [23].

Located in Western Canada, Alberta is one of the most populous provinces in Canada, with approximately 4.4 million people [24]. Approximately 81% of the Alberta population lives in urban centres: 1.2 million people in Calgary [25] and 972,000 in Edmonton [26]. From 2006 to 2011, Calgary and Edmonton have been the fastest growing urban areas [19].

### 2.2. Study Population

The study included all children born in Calgary and Edmonton between 1 April 2005 and 31 March 2010 identified in the Alberta Perinatal Health Program (APHP); a validated clinical perinatal registry of all Alberta births at hospitals or attended by registered midwives at home. The 6-character postal codes of maternal place of residence at delivery reported in the APHP were used to identify children born within the Calgary and Edmonton city limits.

The study methods and population flow diagram of the original retrospective cohort study have been published elsewhere [18]. Briefly, birth cohort data were linked to deidentified, individual-level administrative health data on acute health services (hospitalizations and emergency department [ED] visits) from the Discharge Abstracts Database and the National Ambulatory Care Reporting System. These administrative health datasets register diagnostic information for every episode of acute care using International Classification of Diseases, 10th Revision, enhanced Canadian version (ICD-10-CA) diagnostic codes [27]. For each infant in the birth cohort, we extracted data on all events of acute health services utilization between birth and five years of age—a susceptible time-window for respiratory problems [2]—that had an ICD-10-CA primary diagnostic code indicative of any of the following respiratory conditions: acute bronchiolitis (bronchiolitis/bronchitis) (J20–J21); asthma (J45); croup (J05); influenza (J09–J11); pneumonia (J12–J18); other acute lower respiratory tract infections (J22); and other acute upper respiratory tract infections (J00–J06, except J05). Recurrent wheezing (R06.2) events were merged with asthma or bronchiolitis based on the most prevalent condition after the first wheezing episode. The follow-up period to extract data on respiratory acute care services occurred from 2005 to 2015, with data being censored at date of death or end of follow-up period (i.e., 5 years of age).

### 2.3. Definition of Geographic Areas

We used dissemination areas (DA) as geographic units of analysis to aggregate all respiratory hospitalizations and ED visits occurring between ages 0 to 5 among children born in Calgary and Edmonton. The DA is the smallest (population of ~400 to 700 people) and relatively spatial- and time-stable standard geographic area used by Statistics Canada to disseminate census data [28]. They are larger than postal codes and include approximately 250 households in urban settings. We linked the 6-character postal code of the maternal residence at delivery to the longitude and latitude coordinates from Digital Mapping Technology Inc. (DMTI) Spatial’s Postal Code Suite [29], and to the corresponding DA (*n* = 5357) using the 2006 census geography framework [30]. We performed a vector overlay of 2006-2010 postal code locations [31] with the 2006 DA boundary file to associate postal codes created after 2006 within the 2006 geographic framework.

### 2.4. Explanatory Variable: Spatial Filters

Eigenvector spatial filters (ESF) were used to identify areas in Edmonton and Calgary where a gradient in the prevalence of respiratory services utilization might exist. ESF capture spatial components (i.e., contextual factors operating at different scales) that can be related to a specific health outcome. Briefly, eigenvector spatial filtering is a statistical method to identify spatial patterns in the distribution of a study outcome across a geographical space [32,33,34]. The method is based on eigenvector decomposition of a N × N geographical connectivity matrix to extract orthogonal (uncorrelated) numerical components (eigenvectors). Each eigenvector represents an independent map pattern that captures the latent spatial autocorrelation of a georeferenced variable. Candidate eigenvectors (a subset of the ones related to the health outcome) are linearly combined into a spatial filter. The scores of the spatial filter can then be used as explanatory variable in a regression model.

The ESF solution depends on technical specifications at each step [33,34,35]. We defined a connectivity matrix for the DAs according to the “queen” rule, in which a DA would include neighbourhoods that share boundaries based on a single point (node) or a segment of border limits. Candidate eigenvectors were chosen based on positive spatial correlations using a minimum threshold of 0.25 in the Moran’s index (Moran’s-I) [35]. The Moran’s-I is a dimensionless measure of spatial autocorrelation of the data with values ranging from −1 (usually between −1 and −0.5) to 1 (often slightly larger than 1). Positive values in the Moran’s-I indicate that similar values cluster together in a map, while negative values indicate that dissimilar values are clustered together [36]. A Moran’s-I equal to zero suggests no spatial autocorrelation; whereas values close to one indicate strong spatial autocorrelation. There is no consensus on specific thresholds for the interpretation of categories of weak, moderate, or strong spatial autocorrelation for the Moran’s-I [37]. A subset of eigenvectors statistically related to the study outcome was chosen to form the spatial filter based on a coefficient of determination (R^2^) criterion (see point 2.7). We used the Moran eigenvector spatial filtering software ESF-tool, described by Griffith et al. [34], for extracting the eigenvector’s solution. Additionally, we compared the final results versus the ones obtained from a connectivity matrix based on the “rook” rule; in which polygons are neighbours if they share a segment of border [32]. Results were included in Appendix A.

### 2.5. Study Outcome

Geographic inequalities were defined as spatial differences in the prevalence of respiratory health services utilization among areas defined by the spatial filters. We used smoothed standardized prevalence ratios (smoothed SPR) at the DA level as the primary outcome.

#### Smoothed Standardized Prevalence Ratios (SPR) of Respiratory Health Services Utilization

We calculated the total number of respiratory health services per individual that occurred from birth to 5 years of age as the sum of all ED visits and hospitalizations. The totals were aggregated at postal code level and then at DA level. For each DA, the SPR was calculated using the following formula: SPR = total number of respiratory health services ÷ expected total number of respiratory health services; where the expected total number of respiratory health services = number of singleton live births × overall provincial prevalence of respiratory health services. The SPR was smoothed (smoothed SPR) using empirical Bayes estimators [38], which are based on a Poisson random intercept regression model. Bayes estimators account for unstable prevalence numbers in DAs with low numbers of births and respiratory episodes of care [39]. Briefly, a smoothed SPR equal to 1 indicates that the prevalence of respiratory health services utilization in a particular DA is equal to the expected provincial prevalence; whereas a smoothed SPR > 1 or <1 indicates that the prevalence of respiratory health services utilization in a particular DA is higher or lower, respectively, than the expected provincial prevalence of respiratory health services utilization. We used the programs developed by Rabe-Hesketh and Skrondal [39] for Stata software (version 15.1 [40]) to obtain the smoothed SPR (details for calculations of the smoothed SPR are presented in Appendix A).

### 2.6. Covariates

The relationship between the smoothed SPR and the spatial filter scores was tested in a regression model (multivariable linear) adjusting for other potential covariates operating at spatial level. The idea was to evaluate whether the spatial filter captures a spatial component related to the outcome but independent from the covariates. Area-level socioeconomic status and concentrations of nitrogen dioxide (NO_2_) and fine particulate matter (PM_2.5_) were incorporated as covariates.

We used the 2006-Pampalon material and social deprivation indices as proxy measures of area-level socioeconomic status (SES) [41,42]. The Pampalon indices are area-level composite measures of SES which integrate individual Canadian census data for the population aged 15 and over, excluding First Nations groups. The 2006-Pampalon material and social deprivation indices were derived from mandatory census data. For this reason, the 2006 Pampalon index was used in this research over the 2011-Pampalon index as the latter was built upon a voluntary survey that resulted in a high global nonresponse rate [43]. The material deprivation index is a small-area composite index that integrates data by DA on income, education, and employment; whereas the social deprivation index is composed of marital status, one-person household, and single-parent family’s information. Material and social deprivation indices are reported in quintiles, where Q_1_ and Q_5_ correspond to the least and most deprived groups, respectively.

Area-level concentrations of NO_2_ and PM_2.5_ were derived by Hystad et al. [44] from a national land-use regression model that considered variation in regional and local-scale sources of pollution and incorporated satellite-based estimates, fixed-site monitoring measurements, and geographic predictor variables for the year 2006. This is the only validated land-use regression model of area-level air pollution available for Alberta. We used vector overlay in ArcGIS 10.5 [45] to assign both DA-level Pampalon indices and DA-level NO_2_ and PM_2.5_ concentrations to maternal postal codes at delivery for the study population.

### 2.7. Data Analysis

We used frequencies and percentages to describe baseline demographic characteristics of the study population, their distribution across the material and social deprivation quintiles, and the total number of respiratory health services utilization for each city. We described the DA-level geographic distribution of both material and social deprivation quintiles and the distribution of smoothed SPR in each city using choropleth descriptive maps. Spatial autocorrelation of these variables was reported by the Moran’s-I obtained from the ESF-tool software [34]. Choropleth maps were created using QGIS v3.4.14 software [46].

For the evaluation of geographic inequalities in the prevalence of respiratory services utilization in Edmonton and Calgary, we applied the following analytical steps (Figure 1). First, we evaluated the association between the smoothed SPR and the material and social deprivation quintiles (using Q_1_ as reference), adjusting for PM_2.5_ and NO_2_ in a multivariable linear regression model. From this first model without spatial filters, we verified the presence of spatial autocorrelation in the residuals based on the contiguity of the DAs. Second, we extracted the eigenvectors and selected a set of eigenvectors statistically related to the smoothed SPR. We selected the first eigenvectors that most rapidly increased the R^2^ square of the regression model to achieve a parsimonious model. The set of eigenvectors were linearly combined to form a spatial filter. Third, we evaluated the association between the smoothed SPR and the spatial filter scores adjusting for the material and social deprivation quintiles (using Q_1_ as reference) and area-level concentrations of PM_2.5_ and NO_2_ in a multivariable linear regression model. From this second model with spatial filters, we verified the absence of spatial autocorrelation in the residuals based on the contiguity of the DAs. These three steps were performed using the ESF tool [34]. In the last step, we categorized the spatial filter scores into quintiles to define areas of low (Quintile 1) to high (Quintile 5) respiratory health services utilization. Finally, we recalculated smoothed SPR averages (with 95% confidence intervals [CI]) by geographic areas defined with the spatial filter quintiles to quantify the geographic inequalities using Stata software version 15.1 [40].

## 3. Results

### 3.1. Descriptive Statistics

A total of 111,056 respiratory health services were registered by the 119,909 children of the study population during the follow-up period. Sixty percent (*n* = 66,961) of these episodes of health services use occurred in Calgary and the remaining 40% (*n* = 44,095) in Edmonton. In both Calgary and Edmonton, more than 50% of the respiratory health services occurred during the first two years of age (52% for Calgary, and 55% for Edmonton). Of all children in the study, 5.35% (*n* = 6415) moved from the residential postal code at birth.

The distribution of demographic data across the maternal and social deprivation quintiles and the number of DA for each city are presented in Table 1. In Calgary, 31% of live births occurred in the two most materially deprived quintiles (Q_4_ and Q_5_) while 45% occurred in the corresponding Q_4_ and Q_5_ groups in Edmonton. There was significant heterogeneity in the proportional distribution of live births across maternal and social deprivation quintiles in Calgary and Edmonton (chi-square = 6.9 × 10^3^, *p*-value < 0.01 for the material index; chi-square = 2.1 × 10^3^, *p*-value < 0.01 for the social index).

### 3.2. Calgary

#### 3.2.1. Exploratory Maps

The geographic distribution of material and social deprivation quintiles in Calgary is shown in Figure 2A,B. A high spatial autocorrelation was observed for both indices. The observed Moran’s-I was 0.63 (*p*-value < 0.01) for the material deprivation quintiles and 0.47 (*p*-value < 0.01) for the social deprivation quintiles. The most materially deprived areas were located in the northeast of the city, whereas the most socially deprived areas were located in the central part of the city and along a north-south divide. The geographic distribution of smoothed SPR is shown in Figure 2C. Areas of smoothed SPR > 1 consisted of small local clusters geographically separated among them and scattered across the north-south divide. A low, but significant, spatial autocorrelation index for the smoothed SPR was observed (Moran’s-I = 0.06; *p*-value < 0.01) (Moran’s I scatter plots are shown in Appendix A).

#### 3.2.2. Regression Models for Smoothed-SPR without and with Spatial Filter

The multivariable linear regression model without spatial filter (Model A in Table 2) showed that, compared to the least deprived quintile Q_1_, the smoothed SPR was higher in material deprivation quintiles Q_2_, Q_3_ and Q_5_ and in those in the most socially deprived quintile Q_5_, after adjusting for PM_2.5_ and NO_2_. The model explained 2.4% of the total variance (adjusted-R^2^ = 0.02) but had a significant spatial autocorrelation in the residuals (Moran’s-I of residuals = 0.05, *p*-value < 0.01).

The regression model with a spatial component (Model B in Table 2) incorporated a spatial filter (Figure 2D) based on 14 eigenvectors from a total of 263 eigenvectors describing different patterns of positive spatial autocorrelation. This model was similar to model A in terms of the significance of the independent variables except for Q_4_ in the social deprivation index (which was now statistically significant). The model explained 17% of the total variance (adjusted-R^2^ of 0.17), meaning a 15% improvement in relation to Model A. Additionally, the spatial autocorrelation in residuals was removed (Moran’s-I of residuals = −0.03, *p*-value = 0.60). The inclusion of the spatial filter did not add collinearity to the regression model (variance inflation factor = 1.06), meaning that the spatial filter can be interpreted as an unmeasured spatial explanatory variable independently related to the smoothed SPR (summary statistics and graphs of R^2^ for selected eigenvectors are presented in Appendix A).

#### 3.2.3. Geographic Inequality

The spatial filter quintiles (Figure 2D) were significantly related to the smoothed SPR and suggested a spatial gradient in the prevalence of respiratory health services utilization in early childhood. Overall, the southeast of Calgary had the lowest smoothed SPR, while the highest SPRs were scattered across other city areas. There was an incremental gradient of the smoothed SPR across the zones defined by the spatial filter quintiles (Figure 3A). There was a 1.5-fold increase (or 50% more) in the predicted SPR average between quintile 1 and quintile 5 (SPR = 0.57, CI 0.55 to 0.61 for quintile 1; and SPR = 0.87, CI 0.83 to 0.90 for quintile 5) (Figure 3B).

### 3.3. Edmonton

#### 3.3.1. Exploratory Maps

The geographic distribution of both material and social deprivation quintiles in Edmonton are shown in Figure 4A,B. Spatial autocorrelations were observed for material and social deprivation quintiles. The observed Moran’s-I was 0.56 (*p*-value < 0.01) for the material deprivation index and 0.46 (*p*-value < 0.01) for the social deprivation index. The most materially deprived areas were located in the north and southeast areas of Edmonton, while the most socially deprived areas (Q_4_ and Q_5_) extended across the city. The geographic distribution of smoothed SPR is shown in Figure 4C. Areas of smoothed SPR > 1 were mainly north-central (northeast to west-central); the Moran’s-I autocorrelation was 0.18 (*p*-value < 0.01) (Moran’s I scatter plots are shown in the Appendix A).

#### 3.3.2. Regression Models for Smoothed-SPR without and with Spatial Filter

The multivariable linear regression model without spatial filter (Model A in Table 3) showed that, compared to Q_1_, the smoothed SPR were higher for Q_2_, to Q_5_ for the material deprivation quintiles, and for Q_4_ and Q_5_ for the social deprivation quintiles. The model explained 14% of the total variance (adjusted-R^2^ of 0.14) but had significant spatial autocorrelation in residuals (Moran’s-I of residuals: 0.05, *p*-value < 0.01). The regression model with a spatial component (Model B in Table 3) incorporated a spatial filter (Figure 4D) based on 8 selected eigenvectors from a total of 214. All explanatory variables in this model were statistically significant. The model explained 23% of the total variance (adjusted-R^2^ of 0.23), which meant a 9% improvement in relation to Model A. Additionally, the spatial autocorrelation in residuals was removed (Moran’s-I of residuals: −0.02, *p*-value = 0.67). The inclusion of the spatial filter did not add collinearity (variance inflation factor = 1.1) to the regression model, meaning that the spatial filter can be interpreted as unmeasured spatial explanatory variables independently related to the smoothed SPR (summary statistics and graphs of R^2^ for selected eigenvectors are presented in Appendix A).

#### 3.3.3. Geographic Inequality

The spatial filter quintiles (Figure 4D) were significantly related to the smoothed SPR suggesting a geographic gradient in the prevalence of respiratory health services utilization in early childhood. There was an incremental gradient of the smoothed SPR across zones defined by the spatial filter quintiles (Figure 5A). There was a 1.4-fold increase (or 40% more) in the predicted SPR average between quintiles 1 and 5 (SPR = 0.57, CI 0.54 to 0.61 for quintile 1; and SPR = 0.79, CI 0.76 to 0.82 for quintile 5) (Figure 5B).

## 4. Discussion

This study evaluated geographic inequalities of paediatric respiratory health services utilization (hospitalizations and ED visits) in Calgary and Edmonton, two urban centres with the highest population density in Alberta. We identified a geographic gradient in the distribution of respiratory health inequalities in these two cities. There was a 1.5-fold gap in respiratory health services utilization in Calgary between the areas with the highest and the lowest smoothed SPR, whereas the gap in Edmonton was 1.4-fold. This means that there was 40% to 50% more respiratory health services utilization during early childhood in city areas spatially associated with the highest smoothed SPR compared to zones spatially associated with the lowest smoothed SPR. In Calgary, several small conglomerates of areas scattered across the city had a high demand of respiratory health services, while areas with high demand of respiratory health services in Edmonton followed a regional-cluster spatial distribution.

Results from the regression models indicated that geographic patterns of respiratory health services utilization were only partially explained by the geographic distribution of socioeconomic status in both cities. A nonspatial socioeconomic gradient in health services utilization in Alberta for the respiratory outcomes included in this study has been recently described [18]. That study showed a concentration of paediatric ED visits and hospitalizations for almost all respiratory diseases in the most deprived groups, regardless of the geographic location. The gradient patterns of health inequalities were clearer in the material compared to the social deprivation indices. The substantial increments of the R^2^ values in the models that incorporated the spatial filters indicate that the local geographic patterns of respiratory health inequalities cannot be totally explained by a socioeconomic spatial gradient and that other unmeasured variables act independently of socioeconomic factors. Thereby, the spatial components (spatial filters) that were highly related (statistically) to the geographic distribution of paediatric respiratory health services utilization in both cities, have implications in at least three different aspects of the study.

First, it indicates that unmeasured variables at the neighbourhood level, apart from the material and social status, are associated with children’s respiratory health service utilization. This means that independent missing effects of unobserved neighbourhood contextual characteristics (e.g., access to healthcare facilities, school environments, ethnic disparities, cultural practices, or aeroallergens) should be incorporated to explain geographic inequalities in both cities studied. Unfortunately, we did not have information to identify other contextual factors. A study on socio-spatial polarization in Calgary [47] reported that low income areas in the northeast sector of the city (as shown in our map for material deprivation index) have high concentrations of visible minority immigrants, who may experience difficulties in accessing health care facilities due to limited English language skills. In Edmonton, hotspot areas demanding services for attending mental health, addictions, homelessness, and basic needs [48] are embedded beyond the most socially deprived areas identified in our study (for example, homeless living in wealthy areas). Although additional research is needed to identify other neighbourhood-contextual factors affecting health service utilization of paediatric respiratory conditions, our results support the conclusions from other studies exploring neighbourhood-level factors on health inequalities affecting directly (e.g., housing conditions, aeroallergens [49,50]) or indirectly (e.g., adverse birth outcomes [51]) children’s respiratory outcomes. Interestingly, area-level concentrations of the air pollutants included in this study were not related to the geographic patterns of respiratory health services utilization (in Calgary only after the inclusion of the spatial filter). This is a contrasting result from evidence relating environmental pollutants with respiratory child health [1]. The low spatial variability in NO_2_ and PM_2.5_ concentrations captured from the land use regression models can be a potential explanation for these results (see Appendix A)

Second, the geographic pattern delineated by spatial filter quintiles help to identify the geographic extension of neighbourhood-level factors that could potentially be further investigated. Part of those neighbourhood-level factors can be external factors surrounding areas of high rate in the use of paediatric respiratory healthcare services, for example: allergens (pollen, fungal) or air pollutants that have been identified as cofactors of respiratory diseases [50,52,53]; or social-structural factors limiting the access to health care services including heterogenous geographic distribution of healthcare facilities [54]. In this way, the identification of the spatial extensions related to health outcomes help to empirically conceptualize “neighbourhoods” as social-ecological units beyond the use of administrative boundaries [55] and be of interest for local healthcare authorities.

Finally, the inclusion of spatial filters in the regression models substantially improved model-fitting (i.e., increased adjusted R^2^ in relation to nonspatial models, equally for both Calgary and Edmonton) by removing spatial autocorrelation in residuals. These results improve our understanding of the spatial factors that shape health inequalities [51]. These three characteristics: (1) uncovering missing effects of potential contextual variables; (2) helping to conceptualize the extension of neighbourhood-level factors related to paediatric respiratory outcomes, and; (3) improving statistical inferences, made spatial eigenvector-based analysis a useful technique to incorporate in studies exploring geographic inequalities in health.

### Strengths and Limitations

The strength of the study relies on the use of high-quality administrative databases and a semiparametric spatial technique that captures the relevant spatial configuration of areas (eigenvector-based spatial filter), thereby explaining the geographic distribution of the (smoothed) standardized prevalence ratios. Various modelling approaches have been used by epidemiologists to assess geographic health inequalities: for example, contrasting health indicators among predefined administrative areas [21,56,57], identifying spatial clusters or hotspots as areas of high risk within a particular place [58,59], or the use of multilevel models to analyse small area variation of health outcomes [20,60]. When the spatial patterns of the outcome are complex (e.g., depend on the spatial scale), the characteristics of spatial eigenvector analysis allows us to explore and detect spatial configuration patterns relevant to a specific (health) outcome. This methodology has been successfully used in other research areas to discover spatial/geographic patterns related to human migration [61], species biodiversity [62], and health disparities [51] among other themes. Although its use in geographic health inequalities is, to our knowledge, still limited, it is promising especially when identification of spatial patterns as potential predictors of an outcome is a target [63].

A main limitation in our study relates to the interpretation of the spatial patterns. The spatial patterns result from technical aspects of capturing spatial autocorrelation in spatial analysis. Spatial autocorrelation influences the interpretation of statistical models and is a topic of continuous research in spatial analysis [63]. The ESF approach is one among several techniques that can be used in spatial analysis. We chose ESF because its efficiency in capturing latent spatial autocorrelation at several spatial scales [63]. Spatial models, in general, use different ways to formalize spatial dependence. The definition of a connectivity matrix, either using neighbourhoods sharing administrative boundaries or using distance matrices, will affect spatial dependence [64]. In our study, we defined the connectivity matrix based on areas that were spatially connected based on shared boundaries instead of distances. The use of distance matrices (e.g., Euclidean distances) requires prior information about the spatial processes in question [64]: for example, distances to health facilities, which were unavailable for our study. We used the queen matrix rule to extract our solution and compared to the solution from a rook matrix rule definition in a sensitivity analysis. Under queen connectivity, polygons are neighbours if they share a segment of border or a single node; by contrast, in the rook connectivity, polygons are neighbours if they share a segment of border [32]. The solution from both connectivity definitions was practically the same (detailed results of the sensitivity analysis are included in Appendix A).

Another issue is the selection of eigenvectors for constructing the spatial filter. Several methods have been proposed such as maximization of the multiple regression correlation coefficient, minimization of residual spatial autocorrelation, stepwise selection, among others [62]. Our spatial filter is based, firstly, on positive spatial autocorrelation because we were looking for strongly positively autocorrelated patterns and, secondly, on the set of the first eigenvectors that most rapidly increase the regression multiple correlation coefficient to achieve a parsimonious model. Selecting too many eigenvectors might overcorrect for spatial autocorrelation [62]. For these reasons, interpretation of results could be generalized to the use of a neighbouring connectivity matrix and positive spatial autocorrelation. We acknowledge that the use of distance matrices and/or different eigenvector selection may partially modify the results.

Other study limitations revolve around location, mobility, and model specification. Within urban areas of Canada, the six-character postal codes correspond to an address (e.g., a single building) or group of addresses (e.g., a city block as one side of a street between two intersecting streets) in an urban area [65]. However, residences within newer subdivisions may be less spatially accurate. Mobility of families to other postal codes during the study period introduces misclassification bias. In this regard, we estimated that 5.35% of children reported more than one postal code in hospitalization data. The high percentage (~95%) of children living at the same location from birth to five years of age reduces misclassification bias of location. Other studies in comparable urban centres have found that families change residences around 10–11% during pregnancy [66,67]. Model underspecification is another potential limitation of our regression model as no data were available for indoor air pollution and/or house conditions, which are important predictors of child respiratory health [1]. However, the spatial filters capture unmeasured factors related to the outcomes in a robust way (statistically speaking) by incorporating the spatial dependency into models. The five areas delimited by the spatial filter we used may be related to environmental factors (neighbourhoods with very old houses, or close to air pollution sources), but more research is necessary to identify the factors producing the inequalities we found.

## 5. Conclusions

Geographic inequalities of respiratory health services utilization were identified in Calgary and Edmonton. In Calgary, several small conglomerates of areas dispersed along the city presented high demand of health services, whereas in Edmonton areas with high demand of health services were more in a regional-cluster spatial form. Our results confirmed that other unmeasured factors, beyond socioeconomic status, are key drivers of those inequalities. More research is needed to understand the hidden contextual variables embedded within high-demand areas of paediatric respiratory health services.

## Figures and Tables

**Figure 1 ijerph-17-08973-f001:**
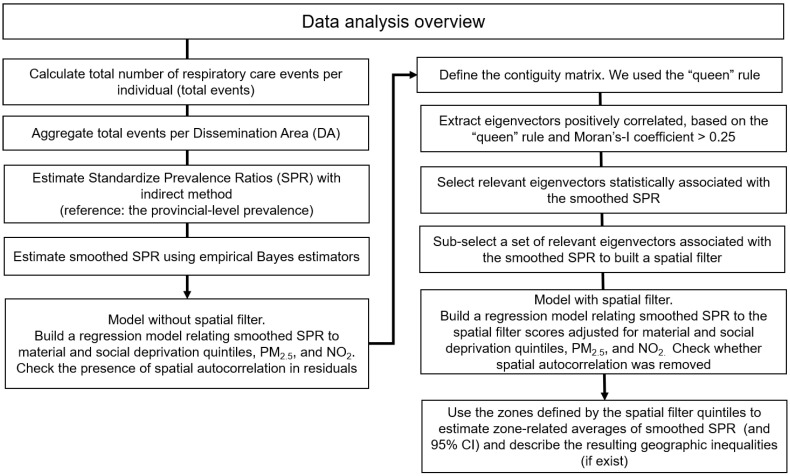
Data analysis overview.

**Figure 2 ijerph-17-08973-f002:**
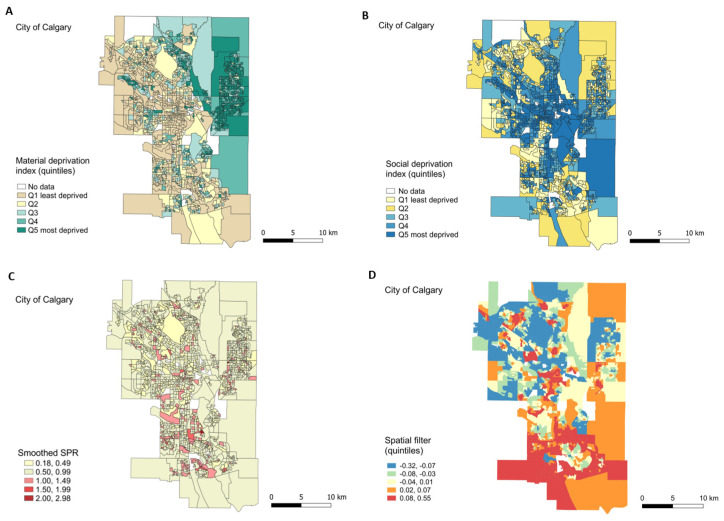
Calgary maps. Geographic distribution of material (**A**) and social deprivation (**B**) quintiles, smoothed SPR (**C**), and spatial filter (**D**). Quintiles were split according to rank values for (**C**,**D**).

**Figure 3 ijerph-17-08973-f003:**
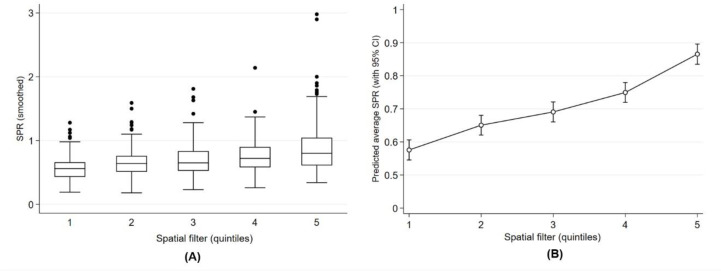
Geographic gradient of smoothed SPR (**A**) and predicted average of SPR (**B**) across Calgary zones defined by the spatial filter quintiles in Calgary.

**Figure 4 ijerph-17-08973-f004:**
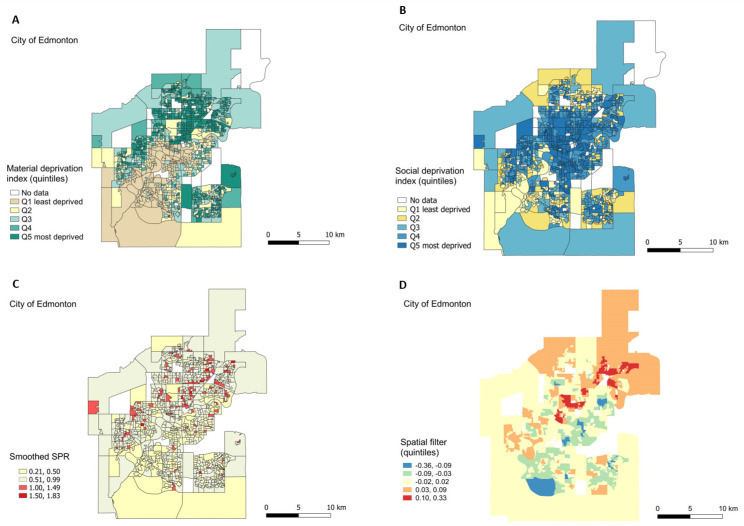
Edmonton maps. Geographic distribution of material (**A**) and social deprivation (**B**) quintiles, smoothed SPR (**C**), and spatial filter (**D**). Quartiles and quintiles split according to rank values for (**C**,**D**), respectively.

**Figure 5 ijerph-17-08973-f005:**
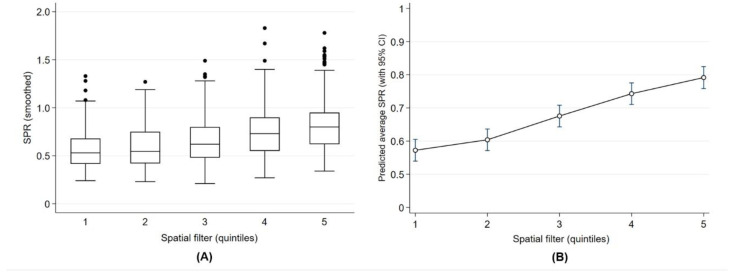
Geographic gradient of smoothed SPR (**A**) and predicted average of SPR (**B**) across Edmonton zones defined by the spatial filter quintiles.

**Table 1 ijerph-17-08973-t001:** Distribution of singleton live births across material and social deprivation quintiles and number of geographic areas (Dissemination Areas: DA) for Edmonton and Calgary.

	Calgary		Edmonton	
	N	%	N	%
Births	70,862	100	49,047	100
Material deprivation quintiles				
Q_1_ (least deprived)	24,313	34	8908	18
Q_2_	13,256	19	6681	14
Q_3_	10,675	15	9667	20
Q_4_	7619	11	11,347	23
Q_5_ (most deprived)	14,110	20	10,945	22
missing	889	1	1499	3
Social deprivation quintiles				
Q_1_ (least deprived)	9682	14	4107	8
Q_2_	16,569	23	9257	19
Q_3_	15,266	22	10,243	21
Q_4_	13,094	18	9865	20
Q_5_ (most deprived)	15,362	22	14,076	29
missing	889	1	1499	3
Number of Dissemination Areas (DA)	1431		1090	

Q = Quintiles.

**Table 2 ijerph-17-08973-t002:** Multivariable linear regression models (without and with spatial filter) on the association between smoothed-SPR and independent variables for Calgary.

	Model A (without Spatial Filter)			Model B (with Spatial Filter)		
Independent Variables	Coefficient	*p*-Value	95% CI	Coefficient	*p*-Value	95% CI
Spatial filter	NA			0.99	0.000	[0.87, 1.12]
Material quintiles						
Q_1_ (least deprived)	Reference			Reference		
Q_2_	0.06	0.009	[0.01, 0.10]	0.04	0.033	[0.00, 0.08]
Q_3_	0.05	0.039	[0.00, 0.09]	0.05	0.023	[0.01, 0.09]
Q_4_	0.01	0.723	[−0.04, 0.05]	0.01	0.700	[−0.03, 0.05]
Q_5_ (most deprived)	0.05	0.016	[0.01, 0.09]	0.07	0.000	[0.04, 0.11]
Social quintiles						
Q_1_ (least deprived)	Reference			Reference		
Q_2_	0.02	0.519	[−0.03, 0.07]	0.02	0.414	[−0.03, 0.06]
Q_3_	0.02	0.502	[−0.03, 0.06]	0.03	0.173	[−0.01, 0.07]
Q_4_	0.02	0.336	[−0.02, 0.07]	0.04	0.047	[0.00, 0.09]
Q_5_ (most deprived)	0.09	0.000	[0.05, 0.14]	0.09	0.000	[0.05, 0.13]
PM_2.5_	0.09	0.002	[0.03, 0.14]	0.01	0.599	[−0.04, 0.07]
NO_2_	0.00	0.079	[−0.01, 0.00]	0.00	0.891	[0.00, 0.01]
constant	0.13	0.506	[−0.24, 0.50]	0.53	0.002	[0.19, 0.88]
	Adjusted R-squared = 0.02			Adjusted R-squared = 0.17		
	AIC = 333.36			AIC = 103.04		
	BIC = 391.07			BIC = 166.00		
	Moran’s-I of residuals: 0.048, *p*-value < 0.01			Moran’s-I of residuals: −0.033, *p*-value = 0.63		

AIC = Akaike Information Criterion. BIC = Bayesian Information. CI = Confidence Interval. NA = Not Applicable. Q = Quintiles.

**Table 3 ijerph-17-08973-t003:** Multivariable linear regression models (without and with spatial filter) on the association between smoothed-SPR and independent variables without and with spatial filter for Edmonton.

	Model A (without Spatial Filter)			Model B (with Spatial Filter)		
Independent Variables	Coefficient	*p*-Value	95% CI	Coefficient	*p*-Value	95% CI
Spatial filter	NA			0.98	0.000	[0.81, 1.15]
Material						
Q_1_ (least deprived)	Reference			Reference		
Q_2_	0.09	0.001	[0.04, 0.15]	0.06	0.024	[0.01, 0.11]
Q_3_	0.10	0.000	[0.05, 0.15]	0.06	0.021	[0.01, 0.11]
Q_4_	0.19	0.000	[0.14, 0.24]	0.12	0.000	[0.07, 0.17]
Q_5_ (most deprived)	0.22	0.000	[0.18, 0.27]	0.15	0.000	[0.10, 0.20]
Social						
Q_1_ (least deprived)	Reference			Reference		
Q_2_	0.04	0.157	[−0.02, 0.10]	0.05	0.063	[−0.01, 0.11]
Q_3_	0.05	0.080	[−0.01, 0.11]	0.05	0.133	[−0.01, 0.10]
Q_4_	0.09	0.001	[0.04, 0.15]	0.10	0.000	[0.05, 0.15]
Q_5_ (most deprived)	0.18	0.000	[0.13, 0.23]	0.18	0.000	[0.13, 0.23]
PM_2.5_	0.01	0.458	[−0.01, 0.03]	0.01	0.905	[−0.02, 0.02]
NO_2_	0.00	0.113	[−0.01, 0.00]	0.00	0.168	[−0.01, 0.00]
Constant	0.47	0.000	[0.30, 0.64]	0.55	0.000	[0.38, 0.71]
	Adjusted R-squared = 0.14			Adjusted R-squared = 0.23		
	AIC = 74.70			AIC = -46.64		
	BIC = 129.08			BIC = 12.68		
	Moran’s-I of residuals: 0.046, *p*-value < 0.01			Moran’s-I of residuals: −0.017, *p*-value = 0.67		

AIC = Akaike Information Criterion. BIC = Bayesian Information. CI = Confidence Interval. NA = Not Applicable. Q = Quintiles.

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
