# Peer review of "Geographic Inequalities of Respiratory Health Services Utilization during Childhood in Edmonton and Calgary, Canada: A Tale of Two Cities"

_ijerph, 2020, doi:10.3390/ijerph17238973_

Round 1

Reviewer 1 Report

The paper is very interesting and well thought-out. Inequalities in children's health are usually analyzed on the basis of mortality statistics or surveys conducted among parents. It also seems that more published data relates to hospital cases than to outpatient clinics. It is clear what knowledge gap this paper has to fill.  In general, the text is difficult to perceive, but it is facilitated by a diagram with subsequent stages of analysis.

I have just a few minor comments, as follows:

  • I suggest rewording the first paragraph of the introduction. The article deals with health inequalities from the perspective of a high developed country. It would be better to focus on morbidity, leaving aside mortality. High mortality due to respiratory diseases is a problem in less developed countries, and there are other underlying causes.
  • Is there any hypothesis behind the comparison between the two cities?
  • The term cross-sectional study/analysis has been repeated several times, in contradiction with the cohort character of research and 5-6 years of observation (even retrospective). Please consider to remove this word in order to clearly define the type of epidemiological study (see line 72 as an example )
  • The data on air pollution in 2006 are a little unclear, since the observation concerned the years 2005-2015. It is difficult to find out which city was more polluted.
  • The discussion is the weakest part of the paper, rather short and focused on methodological issues. I expected more comparisons between two cities and general conclusions. For example, there is a higher R-sq value in Edmonton and a relationship with NO2 and PM2.5 levels revealed only in model A for Calgary.
  • Perhaps it is worth emphasizing in the discussion (limitations) that the data refer to cases and not to children (please confirm if this is true) i.e.  some children are counted repeatedly. Important point because in this respect results based on mortality and morbidity statistics are different.
  • Summary: Geographic inequalities were not completely explained by the spatial distribution. It would be better to write were only in a very small part explained by... because of low R-sq especially in Calgary. 

Editing issues  

7) Please repeat in Table 2 as in Table 1 the labels for Q1 and Q5.

8) In the references, the year of issue should be marked with a bold font.

9) The frequency of repetition of the word 'spatial' is amazing, but there is probably no other option.

Reviewer 2 Report

Thank you for the opportunity to review this manuscript for publication in IJERPH.  The paper investigates respiratory health services utilization amongst young children in two cities in Alberta, Canada. Specifically, it was found that the spatial distribution in access was only partially explained by socioeconomic factors. The paper has a clear research question that is of relevance.  The literature is appropriate, as is the methodology chosen and the conclusions drawn from the results are appropriate.  The paper is also very well written and clear.  Thus, I have no significant concerns.  Below are some very minor suggestions for the authors:

Line 55: Change “off-reserves” to “off reserves”

Line 66-69:  Perhaps add here a brief statement about the way in which way the results of the study might be used practically.

Line 78: Change “populous province” to “populous provinces”

Line 79: Change “million” to “million people”.

Line 95: There is mention on Line 40 about why the 0-5 year age group but it would be good to reiterate here. 

Line 106:  What is the range of areas covered by 1 DA?  It becomes relevance when considering the spatial resolution of the air pollution maps. 

Line 167:  The air pollution concentrations predictions are all based on outdoor levels.  Was any consideration made of indoor levels?  Smoking in particular?  Essentially, one could expect the spatial variability in the outdoor levels as predicted by LUR models to be minimal compared to differences between smoking and non-smoking households.
